# Multi-Label Learning with Pairwise Relevance Ordering

**Ming-Kun Xie and Sheng-Jun Huang**[*]
College of Computer Science and Technology, Nanjing University of Aeronautics and Astronautics
MIIT Key Laboratory of Pattern Analysis and Machine Intelligence, Nanjing, 211106
{mkxie, huangsj}@nuaa.edu.cn

## Abstract

Precisely annotating objects with multiple labels is costly and has become a critical bottleneck in real-world multi-label classification tasks. Instead, deciding the relative order of label pairs is obviously less laborious than collecting exact labels. However, the supervised information of pairwise relevance ordering is less informative than exact labels. It is thus an important challenge to effectively learn with such weak supervision. In this paper, we formalize this problem as a novel learning framework, called multi-label learning with pairwise relevance ordering (PRO). We show that the unbiased estimator of classification risk can be derived with a cost-sensitive loss only from PRO examples. Theoretically, we provide the estimation error bound for the proposed estimator and further prove that it is consistent with respect to the commonly used ranking loss. Empirical studies on multiple datasets and metrics validate the effectiveness of the proposed method.

## 1 Introduction

Multi-label learning (MLL) solves problems where each object is assigned with multiple class labels simultaneously [Zhang and Zhou, 2013]. For example, an image may be annotated with labels *building*, *street* and *person*. The goal of multi-label learning is to train a classification model that can predict all the relevant labels for unseen instances. A large number of recent works have witnessed the great successes that MLL has achieved in many real-world applications, *e.g.*, image annotation [Chen et al., 2019], human attribute recognition [Li et al., 2016], user profiling [Liu et al., 2021], and protein function prediction [Elisseeff and Weston, 2002].

Traditional multi-label learning studies assume that each instance has been precisely annotated with all of its relevant labels. However, in many real-world scenarios, it is difficult and costly to collect the precise annotations. Instead, each instance may be provided with the relative order of label pairs, where each label pair $y \succ y'$ (or $y \prec y'$) indicates that label $y$ is more relevant (or irrelevant) than label $y'$ to instance $\boldsymbol{x}$, i.e., $p(y = 1|\boldsymbol{x}) > p(y' = 1|\boldsymbol{x})$ (or $p(y = 1|\boldsymbol{x}) < p(y' = 1|\boldsymbol{x})$). Generally, deciding the relative order of label pairs would be much easier than collecting the precise annotations and thus less costly. For example, in medical image analysis, only experts with rich experiences can accurately identify the disease for a patient based on the medical image. In contrast, if the question is to decide which of two given diseases is more likely suffered by the patient, then even medical students with basic knowledge may easily provide the answer. While the annotation cost is significantly reduced with pairwise relevance ordering, the learning task becomes more challenging, since the supervised information of pairwise relevance ordering is much less than exact labels.

We formalize this learning problem as a new framework called multi-label learning with pairwise relevance ordering (PRO). Specifically, PRO attempts to learn a classification model from multi-label

---

[*]Correspondence to: Sheng-Jun Huang (huangsj@nuaa.edu.cn).

35th Conference on Neural Information Processing Systems (NeurIPS 2021).

examples with the relative order of label pairs, where each label pair $y \succ y'$ indicates three possible cases: 1) both $y$ and $y'$ are relevant to $\boldsymbol{x}$, i.e., $y = 1, y' = 1$; 2) $y$ is relevant to $\boldsymbol{x}$ while $y'$ is not, i.e., $y = 1, y' = 0$; 3) both $y$ and $y'$ are irrelevant to $\boldsymbol{x}$, i.e., $y = 0, y' = 0$.

PRO is a novel learning framework with significant difference from exiting settings. For example, semi-supervised multi-label learning (SSMLL) learns a classifier by exploiting a few of labeled examples as well as a large number of unlabeled examples [Liu et al., 2006]; multi-label learning with missing labels (MLML) assumes that only a subset of labels are available for each instance [Sun et al., 2010, Yu et al., 2014]; partial multi-label learning (PML) assigns each instance with a candidate label set [Xie and Huang, 2018, Zhang and Fang, 2020]; multi-label learning with noisy labels assumes that multiple class labels may be flipped simultaneously with their respective probabilities [Xie and Huang, 2021a]. However, these frameworks do not consider multi-label examples with pairwise relevance ordering, and cannot be employed to solve PRO problems.

To deal with multi-label data with pairwise relevance ordering, we propose a cost-sensitive loss function for learning a multi-label classifier with empirical risk minimization. Theoretically, we show that the unbiased estimator of classification risk can be derived from only PRO examples if the surrogate loss function satisfies a mild condition, i.e., the symmetric condition. The estimation error bound is established for the unbiased estimator, showing that learning with PRO examples can be multi-label consistent to the commonly used ranking loss. Extensive experimental results on multiple datasets and evaluation metrics demonstrate the practical usefulness of the proposed method.

## 2  Related Works

There are plenty of literature on multi-label learning. As one of the earliest representative methods, Binary Relevance simply decomposes the multi-label learning task into a set of binary classification problems [Zhang and Zhou, 2013]. Nevertheless, such method neglects the label correlations, which are regarded as an essential information for multi-label classification. Therefore, there are many studies trying to learn a multi-label classifier by exploiting the label correlations [Read et al., 2011]. Some of these works focus on the pairwise correlation [Elisseeff and Weston, 2002, Li et al., 2017], while some others consider the high order correlation among all labels [Chen et al., 2019].

To solve SSMLL problems, some works attempt to learn a multi-label classifier based on the graph models [Kong et al., 2011] while some others utilize the low-rank assumption [Jing et al., 2015]. In addition to these methods, the co-training strategy [Zhan and Zhang, 2017] and matrix factorization [Liu et al., 2006] are also employed to solve SSMLL problems.

The pioneering MLML study [Sun et al., 2010] tries to construct a similarity graph for each label and the manifold regularization term is added to recover the missing labels. A linear classifier with the low-rank constraint is proposed to deal with large scale data with missing labels [Yu et al., 2014]. In [Kanehira and Harada, 2016], authors solve the MLML problem by viewing it as a positive-unlabeled learning task. Some other techniques also employ the robust loss [Xu et al., 2019] or the group lasso regularizer [Bucak et al., 2011] to solve MLML problems.

In order to deal with partial-labeled data, the most commonly used strategy is disambiguation [Cour et al., 2011], which recovers ground-truth labeling information for candidate labels. Some methods perform the disambiguation strategy by estimating a confidence for each candidate label [Xie and Huang, 2018, Yu et al., 2018, 2020]. Other methods utilize the decomposition scheme [Sun et al., 2019] or adversarial training [Yan and Guo, 2021]. In [Xie and Huang, 2021b], authors first solve PML problems by considering the generation process of noisy labels. A recent work utilizes the meta disambiguation strategy to deal with partial-labeled data [Xie et al., 2021]. Although it allows noisy labels hidden in the candidate set (*e.g.*, PML) or some labels are missed (*e.g.*, SSMLL and MLML), the aforementioned frameworks consider the supervised information of the label-level, i.e., each of labels is relevant or not, which can be still costly. Instead our proposed PRO framework considers pairwise relevance ordering, which can be much easier for annotators and thus less costly. In [Huang et al., 2015], authors propose a multi-label active learning framework called AURO which queries the relevance ordering between two labels in every iteration. However, different from the proposed PRO framework, AURO directly asks annotators to provide each label pair with one of three possible cases that have been discussed in Section 1, which can be regarded as a stronger supervised information with higher cost. AURO cannot be used to solve the PRO problem. Furthermore, authors consider the pairwise supervision of instances and performing binary classification based on similar paired data

and unlabeled examples [Bao et al., 2018, Shimada et al., 2021, Bao et al., 2020]. Another recent work trains the binary classifier under the supervision of pairwise confidence comparisons [Feng et al., 2021]. These methods focus on binary classification and cannot be directly applied to solve the PRO problem.

# 3 Preliminaries

In this section, before deriving our main results for solving the PRO problem, we introduce some notations and provide the necessary preliminaries.

In traditional multi-label learning task, let $\boldsymbol{x} \in \mathcal{X}$ be a feature vector and $\boldsymbol{y} \subseteq \mathcal{Y}$ its corresponding labels, where $\mathcal{X} = \mathbb{R}^d$ is the feature space and $\mathcal{Y} = \{0, 1\}^q$ is the target space with $q$ possible class labels. Here, $y_j = 1$ indicates the $j$-th label is relevant to the instance; $y_j = 0$, otherwise. Let $\mathcal{D} = \{\boldsymbol{x}_i, \boldsymbol{y}_i\}_{i=1}^n$ be the given training examples, where each example is drawn *i.i.d.* according to the joint distribution $p(\boldsymbol{x}, \boldsymbol{y})$.

In multi-label learning, many loss functions have been proposed to measure the performance of learning algorithms, such as ranking loss, hamming loss, coverage and average precision [Zhang and Zhou, 2013]. Among them, the ranking loss concerns about label pairs that are ordered reversely for an instance, which naturally considers the pairwise label correlation. Given the decision function $\boldsymbol{f} : \mathcal{X} \to \mathbb{R}^d$, the ranking loss can be defined as follows :

$$L(\boldsymbol{f}(\boldsymbol{x}), \boldsymbol{y}) = \sum_{1 \leq j < k \leq q} I(y_j = 1, y_k = 0)\ell(f_k, f_j) + I(y_j = 0, y_k = 1)\ell(f_j, f_k), \quad (1)$$

where

$$\ell(f_j, f_k) = I(f_j > f_k) + \frac{1}{2}I(f_j = f_k). \quad (2)$$

Here, $f_j$ is the $j$-th component of $\boldsymbol{f}(\boldsymbol{x})$ and $I(\cdot)$ is the indicator function, which outputs 1 if the condition holds while outputs 0 otherwise. The goal of multi-label learning tasks is to learn the optimal classifier $\boldsymbol{f}$ by minimizing the following expected classification risk:

$$R(\boldsymbol{f}) = \mathbb{E}_{p(\boldsymbol{x}, \boldsymbol{y})}[L(\boldsymbol{f}(\boldsymbol{x}), \boldsymbol{y}))] = \sum_{\boldsymbol{y} \in \mathcal{Y}} p(\boldsymbol{y})\mathbb{E}_{p(\boldsymbol{x}|\boldsymbol{y})}[L(\boldsymbol{f}(\boldsymbol{x}), \boldsymbol{y}))], \quad (3)$$

$$= \sum_{\boldsymbol{y} \in \mathcal{Y}} p(\boldsymbol{y}) \sum_{1 \leq j < k \leq q} \mathbb{E}_{p(\boldsymbol{x}|y_j=1, y_k=0)}[\ell(f_k, f_j)] + \mathbb{E}_{p(\boldsymbol{x}|y_j=0, y_k=1)}[\ell(f_j, f_k)],$$

$$= \sum_{1 \leq j < k \leq q} \pi_{jk}^{10}\mathbb{E}_{p_{jk}^{10}(\boldsymbol{x})}[\ell(f_k, f_j)] + \pi_{jk}^{01}\mathbb{E}_{p_{jk}^{01}(\boldsymbol{x})}[\ell(f_j, f_k)],$$

where $\pi_{jk}^{10} = p(y_j = 1, y_k = 0)$ (or $\pi_{jk}^{01} = p(y_j = 0, y_k = 1)$) denotes the positive-negative (or negative-positive) label pair prior probability, and $p_{jk}^{10}(\boldsymbol{x}) = p(\boldsymbol{x}|y_j = 1, y_k = 0)$ (or $p_{jk}^{01}(\boldsymbol{x}) = p(\boldsymbol{x}|y_j = 0, y_k = 1)$) denotes the class-conditional probability density of data given the positive-negative (negative-positive) label pair. Accordingly, we define the minimal risk (also called the Bayes risk) as $R^* = \inf_{\boldsymbol{f}} R(\boldsymbol{f})$.

However, the loss function $L$ is highly discontinuous and computationally NP-hard, which often makes the corresponding optimization problem hard to solve [Gao and Zhou, 2013]. In practice, a feasible solution is to consider alternatively a surrogate loss function $\mathcal{L}$ which can be solved efficiently. Accordingly, the $\mathcal{L}$-risk with respect to $p(\boldsymbol{x}, \boldsymbol{y})$ can be defined as:

$$R_{\mathcal{L}}(\boldsymbol{f}) = \mathbb{E}_{p(\boldsymbol{x}, \boldsymbol{y})}[\mathcal{L}(\boldsymbol{f}(\boldsymbol{x}), \boldsymbol{y}))] = \sum_{1 \leq j < k \leq q} \pi_{jk}^{10}\mathbb{E}_{p_{jk}^{10}(\boldsymbol{x})}[\phi(f_j - f_k)] + \pi_{jk}^{01}\mathbb{E}_{p_{jk}^{01}(\boldsymbol{x})}[\phi(f_k - f_j)], \quad (4)$$

where $\phi$ is a surrogate loss function. A common choice is hinge loss $\phi(t) = \max(0, 1 - t)$ in [Elisseeff and Weston, 2002]. Accordingly, we define the minimal $\mathcal{L}$-risk (also called the Bayes $\mathcal{L}$-risk) as $R_{\mathcal{L}}^* = \inf_{\boldsymbol{f}} R_{\mathcal{L}}(\boldsymbol{f})$.

# 4 Learning with PRO

In this section, we first formulate the problem of multi-label learning with pairwise relevance ordering (PRO). Then, the unbiased estimator is proposed for solving the PRO problem. Due to the page limit, most proofs for theorems in Section 4 and Section 5 are provided in the supplementary material.

## 4.1 The PRO Framework

In the PRO framework, each example $x$ is associated with the relevance ordering of $K$ label pairs $\tilde{y} = \{y_j \succ y_j'\}_{j=1}^K$, where $y_j \succ y_j'$ represents the label $y_j$ is more relevant than label $y_j'$, i.e., $p(y_j = 1|x) > p(y_j' = 1|x)$. Note that here $y_j'$ represents one out of $q-1$ labels (except for label $y_j$) and we can further use $y_k$ to denote $y_j'$ for notational simplicity. As discussed in Section 1, $y_j \succ y_k$ indicates three possible cases:1) both $y_j$ and $y_k$ are relevant to $x$, i.e., $y_j = 1, y_k = 1$; 2) $y_j$ is relevant to $x$ while $y_k$ is not, i.e., $y_j = 1, y_k = 0$; 3) both $y_j$ and $y_k$ are irrelevant to $x$, i.e., $y_j = 0, y_k = 0$. The observation tells us that $y_j = 1, y_k = 0$ (or $y_j = 0, y_k = 1$) occurs if and only if $y_j \succ y_k$ (or $y_j \prec y_k$), i.e., the prior probability $p(y_j = 1, y_k = 0|y_j \prec y_k) = 0$ (or $p(y_j = 0, y_k = 1|y_j \succ y_k) = 0$). We can further utilize the prior to calibrate the ordinary risk Eq(3). Therefore, by taking the prior into consideration, the classification risk Eq(3) can be calibrated as:

$$R(f) = \sum_{1 \le j < k \le q} \Pi_{jk}^{10} \mathbb{E}_{\tau_{jk}^{10}(x)}[\ell(f_k, f_j)] + \Pi_{jk}^{01} \mathbb{E}_{\tau_{jk}^{01}(x)}[\ell(f_j, f_k)], \tag{5}$$

where $\Pi_{jk}^{10} = p(y_j = 1, y_k = 0|y_j \succ y_k)$ (or $\Pi_{jk}^{01} = p(y_j = 0, y_k = 1|y_j \prec y_k)$) denotes the calibrated positive-negative (or negative-positive) label pair prior probability, and $\tau_{jk}^{10} = p(x|y_j = 1, y_k = 0, y_j \succ y_k)$ (or $\tau_{jk}^{01} = p(x|y_j = 0, y_k = 1, y_j \prec y_k)$) denotes the calibrated class-conditional probability density of data given the positive-negative (negative-positive) label pair. Accordingly, the $\mathcal{L}$-risk Eq(4) can be calibrated as:

$$R_{\mathcal{L}}(f) = \sum_{1 \le j < k \le q} \Pi_{jk}^{10} \mathbb{E}_{\tau_{jk}^{10}(x)}[\phi(f_j - f_k)] + \Pi_{jk}^{01} \mathbb{E}_{\tau_{jk}^{01}(x)}[\phi(f_k - f_j)]. \tag{6}$$

Our goal is to train a multi-label classifier only based on the observed examples $\tilde{\mathcal{D}} = \{x_i, \tilde{y}_i\}_{i=1}^n$, drawn *i.i.d.* from the distribution $\tilde{p}(x, \tilde{y})$. The expected $\mathcal{L}$-risk with respect to $\tilde{p}(x, \tilde{y})$ can be formulated as follows:

$$\mathbb{E}_{\tilde{p}(x, \tilde{y})}[\mathcal{L}(f(x), \tilde{y})] = \sum_{1 \le j < k \le q} \tilde{\pi}_{jk}^{10} \mathbb{E}_{\tilde{p}_{jk}^{10}(x)}[\phi(f_j - f_k)] + \tilde{\pi}_{jk}^{01} \mathbb{E}_{\tilde{p}_{jk}^{01}(x)}[\phi(f_k - f_j)], \tag{7}$$

where $\tilde{\pi}_{jk}^{10} = p(y_j \succ y_k)$ (or $\tilde{\pi}_{jk}^{01} = p(y_j \prec y_k)$) denotes the positive (or negative) ordering label pair prior probability of PRO examples, and $\tilde{p}_{jk}^{10} = p(x|y_j \succ y_k)$ (or $\tilde{p}_{jk}^{01} = p(x|y_j \prec y_k)$) denotes the class-conditional probability density of PRO examples given the positive (negative) ordering label pair. However, directly solving the estimator Eq.(7) to obtain the classifier usually suffers the over-fitting issue, which makes the classifier fail to obtain a promising generalization performance.

## 4.2 The Proposed Method

In this section, we derive an unbiased risk estimator for solving the PRO problem.

Based on the aforementioned discussions, each label pair $y_j \succ y_k$ indicates three possible cases, which motivates us to derive the following lemma.

**Lemma 1.** *Each PRO example of $\tilde{\mathcal{D}}$ is drawn i.i.d. according a probability distribution with the following class-conditional density:*

$$p(x|y_j \succ y_k) = \Pi_{jk}^{10} p(x|y_j = 1, y_k = 0, y_j \succ y_k) + \pi_{jk}^{11} p(x|y_j = 1, y_k = 1) \tag{8}$$
$$+ \pi_{jk}^{00} p(x|y_j = 0, y_k = 0).$$

Based on the lemma, we derive the following theorem, which obtains the unbiased estimator of the classification risk only from the PRO examples.

**Theorem 1.** *The classification risk Eq.(5) can be equivalently re-written as*

$$R(f) = \sum_{1 \le j < k \le q} \frac{1}{\tilde{\pi}_{jk}^{10}} \left( \tilde{\pi}_{jk}^{10} \mathbb{E}_{\tilde{p}_{jk}^{10}(x)}[\ell(f_k, f_j)] \right) + \frac{1}{\tilde{\pi}_{jk}^{01}} \left( \tilde{\pi}_{jk}^{01} \mathbb{E}_{\tilde{p}_{jk}^{01}(x)}[\ell(f_j, f_k)] \right).$$

*Proof.* According to Eq.(8), we have

$$\mathbb{E}_{\tilde{p}_{jk}^{10}(x)}[\ell(f_k, f_j)] = \Pi_{jk}^{10} \mathbb{E}_{\tau_{jk}^{10}(x)}[\ell(f_k, f_j)] + \pi_{jk}^{11} \mathbb{E}_{p_{jk}^{11}(x)}[\ell(f_k, f_j)] + \pi_{jk}^{00} \mathbb{E}_{p_{jk}^{00}(x)}[\ell(f_k, f_j)].$$

Similarly,

$$\mathbb{E}_{\tilde{p}_{jk}^{01}(\boldsymbol{x})}[\ell(f_j, f_k)] = \Pi_{jk}^{01}\mathbb{E}_{\tau_{jk}^{01}(\boldsymbol{x})}[\ell(f_j, f_k)] + \pi_{jk}^{11}\mathbb{E}_{p_{jk}^{11}(\boldsymbol{x})}[\ell(f_j, f_k)] + \pi_{jk}^{00}\mathbb{E}_{p_{jk}^{00}(\boldsymbol{x})}[\ell(f_j, f_k)].$$

Then, the expected classification risk $R(\boldsymbol{f})$ can be expressed as follows:

$$
\begin{aligned}
R(\boldsymbol{f}) &= \sum_{1 \leq j < k \leq q} \Pi_{jk}^{10}\mathbb{E}_{\tau_{jk}^{10}(\boldsymbol{x})}[\ell(f_k, f_j)] + \Pi_{jk}^{01}\mathbb{E}_{\tau_{jk}^{01}(\boldsymbol{x})}[\ell(f_j, f_k)], \\
&= \sum_{1 \leq j < k \leq q} \mathbb{E}_{\tilde{p}_{jk}^{10}(\boldsymbol{x})}[\ell(f_k, f_j)] + \mathbb{E}_{\tilde{p}_{jk}^{01}(\boldsymbol{x})}[\ell(f_j, f_k)] - \pi_{jk}^{11}\mathbb{E}_{p_{jk}^{11}(\boldsymbol{x})}[\ell(f_k, f_j) + \ell(f_j, f_k)] \\
&\qquad - \pi_{jk}^{00}\mathbb{E}_{p_{jk}^{00}(\boldsymbol{x})}[\ell(f_k, f_j) + \ell(f_j, f_k)], \\
&= \sum_{1 \leq j < k \leq q} \frac{1}{\tilde{\pi}_{jk}^{10}}\left(\tilde{\pi}_{jk}^{10}\mathbb{E}_{\tilde{p}_{jk}^{10}(\boldsymbol{x})}[\ell(f_k, f_j)]\right) + \frac{1}{\tilde{\pi}_{jk}^{01}}\left(\tilde{\pi}_{jk}^{01}\mathbb{E}_{\tilde{p}_{jk}^{01}(\boldsymbol{x})}[\ell(f_j, f_k)]\right) - \pi_{jk}^{11} - \pi_{jk}^{00},
\end{aligned}
$$

where the last equality is due to the fact that $\ell(f_j, f_k) + \ell(f_k, f_j) = 1$ by the definition of $\ell$. $\qquad\square$

However, as discussed in Section 3, it is difficult to optimize the loss function $\ell$ due to its highly discontinuity. To solve the problem, the following corollary tells us that the unbiased estimator of $\mathcal{L}$-risk with respect to $\tilde{p}(\boldsymbol{x}, \tilde{\boldsymbol{y}})$ can be established under a mild condition.

**Corollary 1.** *The $\mathcal{L}$-risk Eq.(6) can be equivalently re-written as*

$$R_{\tilde{\mathcal{L}}}(\boldsymbol{f}) = \sum_{1 \leq j < k \leq q} \frac{1}{\tilde{\pi}_{jk}^{10}}\left(\tilde{\pi}_{jk}^{10}\mathbb{E}_{\tilde{p}_{jk}^{10}(\boldsymbol{x})}[\phi(f_j - f_k)]\right) + \frac{1}{\tilde{\pi}_{jk}^{01}}\left(\tilde{\pi}_{jk}^{01}\mathbb{E}_{\tilde{p}_{jk}^{01}(\boldsymbol{x})}[\phi(f_k - f_j)]\right), \quad (9)$$

*if it holds, for every t, the loss function $\phi$ satisfies*

$$\phi(t) + \phi(-t) = 1, \tag{10}$$

where, for each label pair $y_j, y_k$, $\tilde{\pi}_{jk}^{10}\mathbb{E}_{\tilde{p}_{jk}^{10}(\boldsymbol{x})}[\phi(f_j - f_k)]$ (or $\tilde{\pi}_{jk}^{01}\mathbb{E}_{\tilde{p}_{jk}^{01}(\boldsymbol{x})}[\phi(f_k - f_j)]$) is the expected $\mathcal{L}$-risk with respect to $\tilde{p}(\boldsymbol{x}, \tilde{\boldsymbol{y}})$ and can be directly estimated from PRO training examples with suitable surrogate loss functions. It is noteworthy that the symmetric condition Eq(10) has been widely used in other weakly supervised learning frameworks [Du Plessis et al., 2014, Ishida et al., 2017].

With the theorem, we can train a multi-label classifier by minimizing the empirical approximation of $R_{\tilde{\mathcal{L}}}(\boldsymbol{f})$ from PRO examples as follows:

$$\widehat{R}_{\tilde{\mathcal{L}}}(\boldsymbol{f}) = \frac{1}{n}\sum_{i=1}^{n} \tilde{\mathcal{L}}(\boldsymbol{f}(\boldsymbol{x}), \tilde{\boldsymbol{y}}), \tag{11}$$

where,

$$\tilde{\mathcal{L}}(\boldsymbol{f}(\boldsymbol{x}), \tilde{\boldsymbol{y}}) = \sum_{1 \leq j < k \leq q} \frac{1}{\tilde{\pi}_{jk}^{10}}I(y_{ij} \succ y_{ik})\phi(f_j(\boldsymbol{x}_i) - f_k(\boldsymbol{x}_i)) + \frac{1}{\tilde{\pi}_{jk}^{01}}I(y_{ij} \prec y_{ik})\phi(f_k(\boldsymbol{x}_i) - f_j(\boldsymbol{x}_i)).$$
$$\tag{12}$$

Here, it is worthy noting that in contrast to previous cost-sensitive methods [Du Plessis et al., 2014], which often requires extra assumptions or sophisticated techniques to obtain the cost coefficients, the $\tilde{\pi}_{jk}^{10}$ (or $\tilde{\pi}_{jk}^{01}$) can be directly estimated from the observed PRO training data.

## 5 Theoretical Analysis

In this section, we provide the estimation error bound for the proposed unbiased estimator and further prove that it is consistent with respect to ranking loss.

Let $\boldsymbol{\sigma} = \{\sigma_1, ..., \sigma_n\}$ be $n$ Rademacher variables with $\sigma_i$ independently uniform variable taking value in $\{-1, +1\}$. Then, the Rademacher complexity with respect to function class $\mathcal{F}$ and loss function $\mathcal{L}$ can be formulated as follows:

$$\mathcal{R}_n(\mathcal{L} \circ \mathcal{F}) = \mathbb{E}_{\boldsymbol{x}, \boldsymbol{y}, \boldsymbol{\sigma}}\left[\sup_{\boldsymbol{f} \in \mathcal{F}} \frac{1}{n}\sum_{i=1}^{n} \sigma_i \mathcal{L}(\boldsymbol{f}(\boldsymbol{x}_i), \boldsymbol{y}_i)\right]. \tag{13}$$

Based on the definition, we can establish the following lemma.

**Lemma 2.** *Let $\mathcal{R}_n(\tilde{\mathcal{L}} \circ \mathcal{F})$ be the Rademacher complexity of the loss function $\tilde{\mathcal{L}}$ and function class $\mathcal{F}$ over $\tilde{\mathcal{D}}$ of $n$ training examples drawn from $\tilde{p}(\boldsymbol{x}, \tilde{\boldsymbol{y}})$, which can be defined as*

$$\mathcal{R}_n(\tilde{\mathcal{L}} \circ \mathcal{F}) = \mathbb{E}_{\tilde{\mathcal{D}}} \mathbb{E}_{\boldsymbol{\sigma}} \left[ \sup_{\boldsymbol{f} \in \mathcal{F}} \frac{1}{n} \sum_{i=1}^{n} \sigma_i \tilde{\mathcal{L}}(\boldsymbol{f}(\boldsymbol{x}_i), \tilde{\boldsymbol{y}}_i) \right].$$

*Then,*

$$\mathcal{R}_n(\tilde{\mathcal{L}} \circ \mathcal{F}) \le 2K\pi C_\phi \mathcal{R}_n(\mathcal{F}),$$

*where $\pi = \max_{j,k} \frac{1}{\pi_{jk}^{10}}, \forall j, k \in [q]$ and $C_\phi$ is the Lipschitz constant of $\phi$.*

Based on Lemma 2, we can establish the uniform deviation bounds of $\widehat{R}_{\tilde{\mathcal{L}}}(\boldsymbol{f})$ as follows:

**Lemma 3.** *For the loss function $\phi$ bounded by $\Theta$ and any $\delta > 0$, with the probability at least $1 - \delta$, we have*

$$\max_{\boldsymbol{f} \in \mathcal{F}} |\widehat{R}_{\tilde{\mathcal{L}}}(\boldsymbol{f}) - R_{\tilde{\mathcal{L}}}(\boldsymbol{f})| \le 4K\pi C_\phi \mathcal{R}_n(\mathcal{F}) + K\pi\Theta\sqrt{\frac{\ln \frac{1}{\delta}}{2n}}.$$

Based on Lemma 3, we can derive the estimation error bound as follows, which further shows that learning from PRO examples can be multi-label consistent with respect to ranking loss.

**Theorem 2.** *For any $\delta > 0$, with probability at least $1 - \delta$, we have*

$$R_{\mathcal{L}}(\hat{\boldsymbol{f}}) - \min_{\boldsymbol{f} \in \mathcal{F}} R_{\mathcal{L}}(\boldsymbol{f}) \le 8K\pi C_\phi \mathcal{R}_n(\mathcal{F}) + 2K\pi\Theta\sqrt{\frac{\ln \frac{1}{\delta}}{2n}}.$$

*where $\hat{\boldsymbol{f}}$ is trained by minimizing $\widehat{R}_{\tilde{\mathcal{L}}}(\boldsymbol{f})$.*
*Furthermore, if $\phi$ is a differential and non-increasing function with $\phi'(0) < 0$ and $\phi(t) + \phi(-t) = 2\phi(0)$, then learning from PRO data with the modified loss function $\tilde{\mathcal{L}}$ Eq.(12) is consistent w.r.t ranking loss, i.e., there exists a non-negative concave function $\xi$ with $\xi(0) = 0$, such that*

$$R(\hat{\boldsymbol{f}}) - R^* \le \xi(R_{\mathcal{L}}(\hat{\boldsymbol{f}}) - R_{\mathcal{L}}^*).$$

Theorem 2 tells us that learning from PRO examples is consistent with respect to ranking loss. As $n \to \infty$, if $R_{\mathcal{L}}(\hat{\boldsymbol{f}}) = R_{\mathcal{L}}^*$, then we have the consistency: $R(\hat{\boldsymbol{f}}) = R(\boldsymbol{f}^*)$, since $\mathcal{R}_n(\mathcal{F}) \to 0$ for all parametric models with a bounded norm such as deep networks trained with weight decay [Lu et al., 2018]. Based on the above discussion, Sigmoid loss $\phi(t) = \frac{1}{1+e^t}$ is a suitable surrogate loss function in our case, since it satisfies the symmetric condition Eq.(10) and meanwhile has been proven to be consistent with respect to ranking loss [Gao and Zhou, 2013].

# 6 Experiment

In this section, to validate the effectiveness of the proposed method, we perform the experiments on varied datasets with multiple evaluation metrics.

## 6.1 Experimental Settings

**Datasets** We evaluate our method on five multi-label datasets: Multi-MNIST[2] [Finn et al., 2017], Multi-Kuzushiji-MNIST (Multi-KMNIST for short), Multi-Fashion-MNIST [3] (Multi-FMNIST for short), VOC2007 [4] [Everingham et al., 2010] and MSCOCO [5] [Lin et al., 2014]. For three Multi-MNIST-style datasets, we randomly sample 6,000 images for training and 4,000 images for testing. VOC2007 contains 9,963 images for 20 object categories, which are divided into train, val and test sets. Following [Chen et al., 2019, 2018], we use the *trainval* set to train the models, and evaluate the

---

[2]See `https://github.com/shaohua0116/MultiDigitMNIST` for Multi-MNIST.

[3]Similar to Multi-MNIST, we construct Multi-Kuzushiji-MNIST and Multi-Fashion-MNIST for two commonly used datasets Kuzushiji-MNIST and Fashion-MNIST, repsectively.

[4]See `http://host.robots.ox.ac.uk/pascal/VOC/voc2007/` for VOC2007.

[5]See `https://cocodataset.org` for MSCOCO.

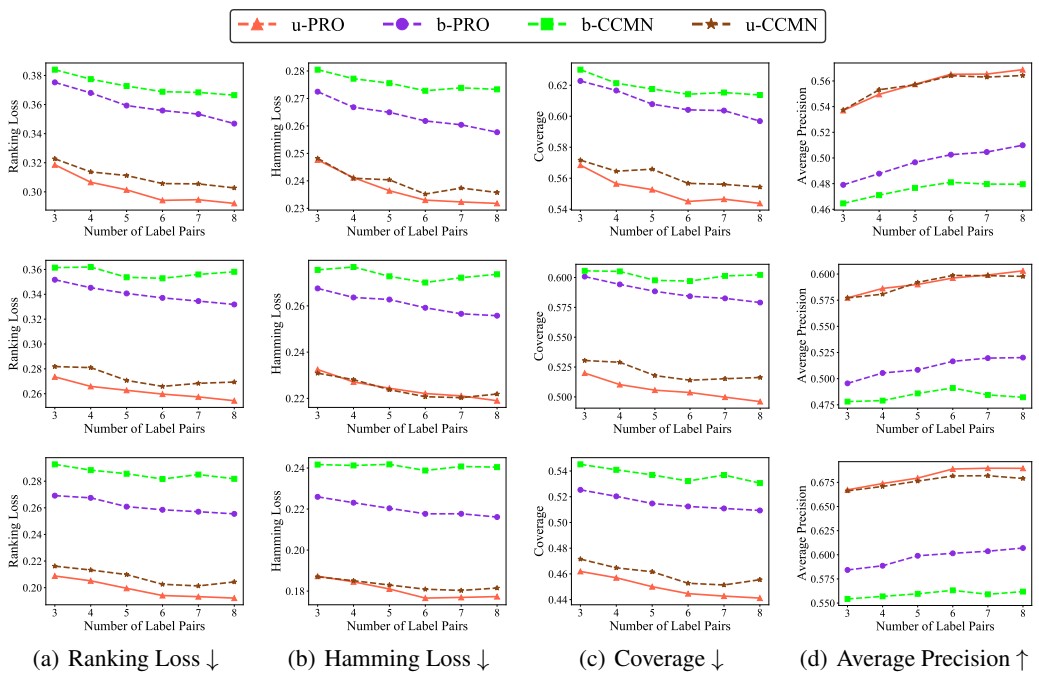

|  |  |  |  |
|---|---|---|---|
| (a) Ranking Loss ↓ | (b) Hamming Loss ↓ | (c) Coverage ↓ | (d) Average Precision ↑ |

Figure 1: Comparison results with varying number of label pairs on Multi-MNIST, Multi-KMNIST and Multi-FMNIST.

performance on the test set. MSCOCO contains 82,081 images as the training set and 40,504 images as the validation set. We randomly sample 20,000 images from the training set for training and 10,000 images from the validation set for testing. For each dataset, we randomly sample $K$ pairs of labels and assign their relevance ordering, where $K$ varies among $\{3, 4, 5, 6, 7, 8\}$ for Multi-MNIST-style datasets, $\{6, 8, 10, 12, 14, 16\}$ for VOC2007 and $\{5, 10, 15, 20, 25, 30\}$ for MSCOCO. In particular, in the case that two labels are both positive (or negative), we decide their pairwise relevance ordering randomly, i.e., one out of two labels is randomly sampled to be more relevant to the other one. For each dataset, we repeat experiments five times and report their averaging performances.

**Metrics**    We evaluate the performance of the proposed method based on multiple standard multi-label criterion: ranking loss, hamming loss, coverage and average precision. For ranking loss, hamming loss and coverage, the smaller value, the better performance; for average precision, the larger value, the better performance. The detail of these criterion can be found in [Zhang and Zhou, 2013].

**Methods**    Under the PRO framework, the proposed method that minimizes $\widehat{R}_{\tilde{\mathcal{L}}}(\boldsymbol{f})$ in Eq.(11) with Sigmoid loss function is denoted by **Unbiased-PRO** (**u-PRO** for short). We compare with the baseline: **Biased-PRO** (**b-PRO** for short), which attempts to minimize the empirical approximation of the biased classification risk in Eq(7) with hinge loss function. Note that PRO is new learning framework, and there is no method can be directly applied to PRO problems. We employ a recently proposed framework called CCMN [Xie and Huang, 2021a] to transform the PRO problem into a MLL problem with class-conditional multi-label noise (CCMN) by regarding $y$ as the positive label while $y'$ as the negative label for each label pair $y \succ y'$. And we compare with the following methods: **Biased-CCMN** (**b-CCMN** for short), which directly learns a multi-label classifier with noisy labels; **Unbiased-CCMN** (**u-CCMN**), which employs the unbiased estimator proposed in [Xie and Huang, 2021] to solve the transformed CCMN problem. Note that for u-CCMN, the true noise rates (the probability of the positive (negative) label flipped into the negative (positive) one) are given in experiments.

**Implementation**    For experiments on Multi-MNIST-style datasets, we train a linear model by using Adam [Kingma and Ba, 2015] optimizer with learning rate of 0.001. We added an $\ell_2$-regularization term, with the regularization parameter of 0.0001. For experiments on VOC2007 and MSCOCO, we use an Alexnet [Krizhevsky et al., 2012] and a Resnet-18 [He et al., 2016] pre-trained with the

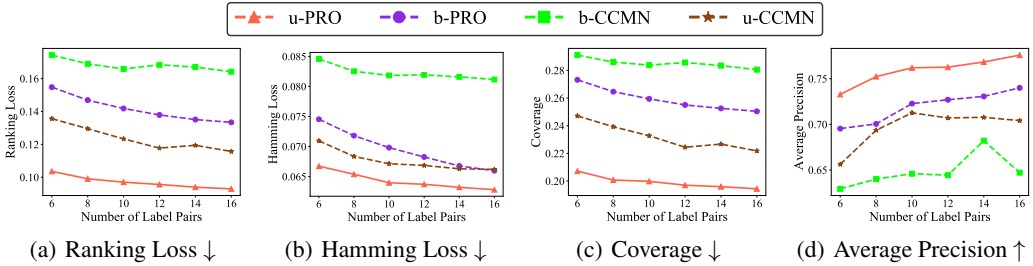

Figure 2: Comparison results with varying number of label pairs for each instance on VOC2007.

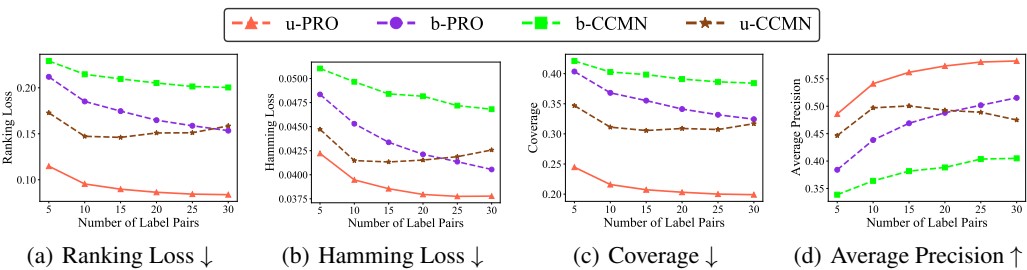

Figure 3: Comparison results with varying number of label pairs for each instance on MSCOCO.

ILSVRC2012 dataset on Pytorch platform [Paszke et al., 2019]. The Alexnet and Resnet-18 are trained by using stochastic gradient descent (SGD) with learning rate of 0.0001. An $\ell_2$-regularization term is added with the regularization parameter of 0.0001. The batch size for all datasets is set as 200. All the experiments are conducted on GeForce RTX 2080 GPUs

## 6.2 Performance Comparison

Figure 1 illustrates the performance curve of each comparing method as the number of label pairs for each instance increases in terms of four evaluation metrics on three Multi-MNIST-style datasets. As shown in the figures, we can obtain the following observations: 1) b-PRO and b-CCMN achieve the worst performances, which indicates neither directly minimizing the empirical biased classification risk nor simply conducting binary classification transformation can solve PRO problems, since these two methods may suffer from over-fitting issues due to the biasedness of the risk estimation. 2) Compared to b-CCMN method, u-CCMN achieves a promising performance in most cases. This observation demonstrates that CCMN framework is effective for tackling PRO problems in some extent. 3) Our proposed unbiased-PRO method achieves the best performance in almost all cases and significantly outperforms u-CCMN. It is worthy noting that u-CCMN utilizes the true noise rates which are usually unavailable in practice, and thus the superiority of the proposed method would be more significant in real-world setting.

Figure 2 and Figure 3 illustrate the performance curve of each comparing method as the number of label pairs for each instance increases in terms of four evaluation metrics on VOC2007 and MSCOCO, respectively. From the figures, it can be observed that our proposed u-PRO method achieves the best performance in all cases. It seem that u-CCMN performs unstable on VOC2007. It even obtains worse results than the baseline b-PRO in terms of hamming loss and average precision. One possible reason is that u-CCMN suffers from the over-fitting issue when the complex model is used (in the experiments, Alexnet is used for VOC2007). These results convincingly validate that the proposed unbiased estimator can effectively solve PRO problems.

## 6.3 Ablation Study

In this section, we conduct some ablation experiments to provide empirical validations for the theoretical analysis proposed in the paper.

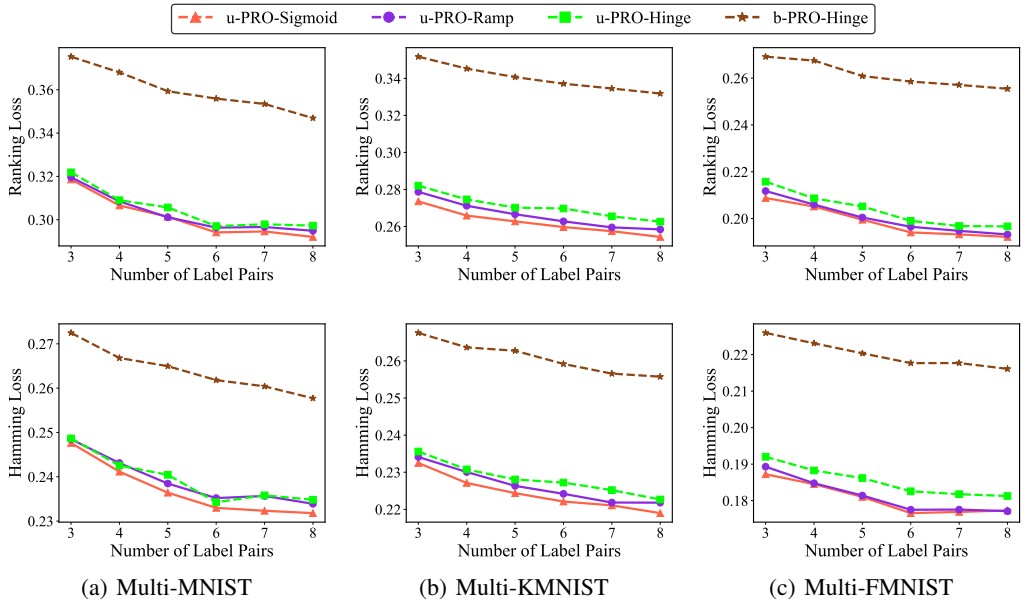

Figure 4: Comparison results with varying number of label pairs for each instance on Multi-MNIST, Multi-KMNIST and Multi-FMNIST in terms of ranking loss and hamming loss.

We first examine the unbiasedness for the proposed estimator. Based on the discussion in Section 4.2, we disclose that the unbiased estimator is composed of two components, i.e., the cost-sensitive estimator Eq.(9) and the symmetric surrogate loss function which satisfies Eq.(10). Here, u-PRO-Sigmoid and u-PRO-Ramp represent the empirical cost-sensitive estimator Eq.(11) with Sigmoid and Ramp losses, respectively, which both satisfy the symmetric condition. U-PRO-Hinge and b-PRO-Hinge represent the cost-sensitive estimator Eq.(11) and biased estimator Eq.(7) with hinge loss, which does not satisfy the symmetric condition.

Due to the page limit, Figure 4 only report the performance curves of these four estimators in terms of ranking loss and hamming loss on Multi-MNIST, Multi-KMNIST and Multi-FMNIST datasets. From the figures, we can obtain following observations: 1) u-PRO-Sigmoid and u-PRO-Ramp achieve the better performances than u-PRO-Hinge and b-PRO-Hinge in almost all cases, which indicates both two components, i.e., the cost-sensitive estimator and the symmetric surrogate loss function, certainly contribute to obtain the unbiased estimator for solving the PRO problem; 2) U-PRO-Hinge outperforms b-PRO-Hinge with significant superiority, which indicates that the cost-sensitive estimator plays an important role in achieving unbiased risk estimation. The observation tells that even without the symmetric surrogate loss function, we can obtain a promising result by utilizing the cost-sensitive estimator in practice. Finally, from the Figure 4, it can be observe that u-PRO-Sigmoid generally achieve better performance than u-PRO-Ramp in terms of ranking loss, which provides an empirical validation of Theorem 2, since Sigmoid loss has been proven to be consistent with respect to ranking loss while Ramp loss is not [Gao and Zhou, 2013].

# 7 Conclusion

In this paper, we study the problem of multi-label classification with pairwise relevance ordering, where each instance is assigned with the relative order of label pairs. To solve PRO problems, we propose an empirical estimator of classification risk based on a cost-sensitive loss. Theoretically, we shows that the proposed estimator can be in an unbiased fashion if the surrogate loss function satisfies the symmetric condition. We derive the estimation error bound for the proposed method, and further prove that learning from PRO examples with the proposed unbiased estimator is consistent with respect to ranking loss. Finally, we experimentally examine the effectiveness of the proposed method on multiple datasets and evaluation metrics. In the future, we will study PRO problems by considering the data generation process.

## 8 Acknowledgments

This research was supported by the National Key R&D Program of China (2020AAA0107000), Natural Science Foundation of Jiangsu Province of China (BK20211517), and NSFC (62076128, 61732006).

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
