# Supplementary Material for Multi-Label Learning with Pairwise Relevance Ordering

## 1 Proof of Lemma 1

Based on the discussion of the main paper, we have

$$
\begin{align}
p(\boldsymbol{x}|y_j \succ y_k) &= p(y_j = 1, y_k = 0|y_j \succ y_k)p(\boldsymbol{x}|y_j = 1, y_k = 0, y_j \succ y_k) \tag{1}\\
&+ p(y_j = 1, y_k = 1|y_j \succ y_k)p(\boldsymbol{x}|y_j = 1, y_k = 1, y_j \succ y_k) \tag{2}\\
&+ p(y_j = 0, y_k = 0|y_j \succ y_k)p(\boldsymbol{x}|y_j = 0, y_k = 0, y_j \succ y_k) \tag{3}\\
&= p(y_j = 1, y_k = 0|y_j \succ y_k)p(\boldsymbol{x}|y_j = 1, y_k = 0, y_j \succ y_k) \tag{4}\\
&+ p(y_j = 1, y_k = 1)p(\boldsymbol{x}|y_j = 1, y_k = 1) \tag{5}\\
&+ p(y_j = 0, y_k = 0)p(\boldsymbol{x}|y_j = 0, y_k = 0) \tag{6}
\end{align}
$$

The equalities of the second term and third term (i.e., line (2) equals line (5) and line (3) equals line (6)) are based on the assumption that for each label pair $(y_j = 1, y_k = 1)$ (or $(y_j = 0, y_k = 0)$), the relevance ordering of the label pair is determined uniformly. The assumption can be expressed as $p(y_j = 1, y_k = 1|y_j \succ y_k) = p(y_j = 1, y_k = 1|y_k \succ y_j) = p(y_j = 1, y_k = 1)$ (or $p(y_j = 0, y_k = 0|y_j \succ y_k) = p(y_j = 0, y_k = 0|y_k \succ y_j) = p(y_j = 0, y_k = 0)$). $\qquad\square$

## 2 Proof of Corollary 1

Similar to Theorem 1, we have:

$$
\begin{align}
R_{\mathcal{L}}(\boldsymbol{f}) &= \sum_{1 \le j < k \le q} \Pi_{jk}^{10}\mathbb{E}_{\tau_{jk}^{10}(\boldsymbol{x})}[\phi(f_j - f_k)] + \Pi_{jk}^{01}\mathbb{E}_{\tau_{jk}^{01}(\boldsymbol{x})}[\phi(f_k - f_j)]\\
&= \sum_{1 \le j < k \le q} \mathbb{E}_{\tilde{p}_{jk}^{10}(\boldsymbol{x})}[\phi(f_j - f_k)] + \mathbb{E}_{\tilde{p}_{jk}^{01}(\boldsymbol{x})}[\phi(f_k - f_j)] - \pi_{jk}^{11}\mathbb{E}_{p_{jk}^{11}(\boldsymbol{x})}[\phi(f_j - f_k) + \phi(f_k - f_j)]\\
&\qquad - \pi_{jk}^{00}\mathbb{E}_{p_{jk}^{00}(\boldsymbol{x})}[\phi(f_j - f_k) + \phi(f_k - f_j)]\\
&= \sum_{1 \le j < k \le q} \frac{1}{\tilde{\pi}_{jk}^{10}}\left(\tilde{\pi}_{jk}^{10}\mathbb{E}_{\tilde{p}_{jk}^{10}(\boldsymbol{x})}[\phi(f_j - f_k)]\right) + \frac{1}{\tilde{\pi}_{jk}^{01}}\left(\tilde{\pi}_{jk}^{01}\mathbb{E}_{\tilde{p}_{jk}^{01}(\boldsymbol{x})}[\phi(f_k - f_j)]\right) - \pi_{jk}^{11} - \pi_{jk}^{00}\\
&= R_{\tilde{\mathcal{L}}}(\boldsymbol{f})
\end{align}
$$

where the forth equality is due to the condition $\phi(f_j - f_k) + \phi(f_k - f_j) = 1$ in the theorem statement. $\qquad\square$

## 3 Proof of Lemma 2

Recall that

$$
\mathcal{L}(\boldsymbol{f}(\boldsymbol{x}), \tilde{\boldsymbol{y}}) = \sum_{1 \le j < k \le q} \frac{1}{\tilde{\pi}_{jk}^{10}}I[y_{ij} \succ y_{ik}]\phi(f_k - f_j) + \frac{1}{\tilde{\pi}_{jk}^{01}}I[y_{ik} \succ y_{ij}]\phi(f_j - f_k) \tag{7}
$$

Then, we have

$$\mathcal{R}_n(\tilde{\mathcal{L}} \circ \mathcal{F}) \tag{8}$$

$$= \mathbb{E}_{\tilde{\mathcal{D}}} \mathbb{E}_\sigma \left[ \sup_{f_1,\dots,f_q \in \mathcal{F}} \frac{1}{n} \sum_{i=1}^n \sigma_i \sum_{1 \le j < k \le q} \frac{1}{\widetilde{\pi}_{jk}^{10}} I[y_{ij} \succ y_{ik}] \phi(f_k - f_j) + \frac{1}{\widetilde{\pi}_{jk}^{01}} I[y_{ik} \succ y_{ij}] \phi(f_j - f_k) \right]$$

$$= \mathbb{E}_{\mathcal{X}} \mathbb{E}_\sigma \left[ \sup_{f_1,\dots,f_q \in \mathcal{F}} \frac{1}{n} \sum_{\boldsymbol{x}_i \in \mathcal{X}} \sigma_i \sum_{1 \le j < k \le q} \frac{1}{\widetilde{\pi}_{jk}^{10}} I[y_{ij} \succ y_{ik}] \phi(f_k - f_j) + \frac{1}{\widetilde{\pi}_{jk}^{01}} I[y_{ik} \succ y_{ij}] \phi(f_j - f_k) \right]$$

$$\le \sum_{1 \le j < k \le q} \mathbb{E}_{\mathcal{X}} \mathbb{E}_\sigma \left[ \sup_{f_1,\dots,f_q \in \mathcal{F}} \frac{1}{n} \sum_{\boldsymbol{x}_i \in \mathcal{X}} \sigma_i (\frac{1}{\widetilde{\pi}_{jk}^{10}} I[y_{ij} \succ y_{ik}] \phi(f_k - f_j) + \frac{1}{\widetilde{\pi}_{jk}^{01}} I[y_{ik} \succ y_{ij}] \phi(f_j - f_k)) \right] \tag{9}$$

Sequentially, let $(y, y')$ be the current label pair to be cumulated, then, according to Talagrand's contraction lemma [Ledoux and Talagrand, 2013], we have

$$\mathbb{E}_{\mathcal{X}} \mathbb{E}_\sigma \left[ \sup_{f_y, f_{y'} \in \mathcal{F}} \frac{1}{n} \sum_{\boldsymbol{x}_i \in \mathcal{X}} \sigma_i (\frac{1}{\widetilde{\pi}_{jk}^{10}} I[y_{ij} \succ y_{ik}] \phi(f_k - f_j) + \frac{1}{\widetilde{\pi}_{jk}^{01}} I[y_{ik} \succ y_{ij}] \phi(f_j - f_k)) \right]$$

$$\le \pi C_\phi \mathbb{E}_{\mathcal{X}} \mathbb{E}_\sigma \left[ \sup_{f_y, f_{y'} \in \mathcal{F}} \frac{1}{n} \sum_{\boldsymbol{x}_i \in \mathcal{X}} \sigma_i (I(y \succ y')(f_{y'}(\boldsymbol{x}_i) - f_y(\boldsymbol{x}_i)) + I(y \prec y')(f_y(\boldsymbol{x}_i) - f_{y'}(\boldsymbol{x}_i))) \right]$$

$$\le \pi C_\phi \mathbb{E}_{\mathcal{X}} \mathbb{E}_\sigma \left[ \sup_{f_y \in \mathcal{F}} \frac{1}{n} \sum_{\boldsymbol{x}_i \in \mathcal{X}} \sigma_i f_y(\boldsymbol{x}_i) \right] + C_\phi \mathbb{E}_{\mathcal{X}} \mathbb{E}_\sigma \left[ \sup_{f_{y'} \in \mathcal{F}} \frac{1}{n} \sum_{\boldsymbol{x}_i \in \mathcal{X}} \sigma_i f_{y'}(\boldsymbol{x}_i) \right]$$

$$= 2\pi C_\phi \mathcal{R}_n(\mathcal{F})$$

where $f_y$ represents the classifier for class label $y$. Then, with Eq.(9), it is easy to prove that

$$\mathcal{R}_n(\tilde{\mathcal{L}} \circ \mathcal{F}) \le 2K\pi C_\phi \mathcal{R}_n(\mathcal{F}).$$

$\square$

## 4 Proof of Lemma 3

Since both two directions can be proved in the same way, we consider one single direction $\sup_{f_1,\dots,f_q \in \mathcal{F}} (\widehat{R}_{\tilde{\mathcal{L}}}(\boldsymbol{f}) - R_{\tilde{\mathcal{L}}}(\boldsymbol{f}))$. Note that the change in $\boldsymbol{x}_i$ leads to a perturbation of at most $\frac{K\mu\Theta}{n}$ by replacing a single point $(\boldsymbol{x}_i, \tilde{\boldsymbol{y}}_i)$ with $(\boldsymbol{x}'_i, \tilde{\boldsymbol{y}}'_i)$, since the change in any $\tilde{y}_j^{(i)}$ leads to a perturbation as $\frac{1}{n} \left| \frac{\phi(f_k - f_j)}{\widetilde{\pi}_{jk}^{10}} \right| \le \frac{\pi\Theta}{n}$, where $\pi = \max_{j,k \in [q]} \frac{1}{\widetilde{\pi}_{jk}^{10}}$. By using McDiarmid' inequality [Mohri et al., 2018] to the single-direction uniform deviation $\sup_{f_1,\dots,f_k \in \mathcal{F}} \widehat{R}_{\tilde{\mathcal{L}}}(\boldsymbol{f}) - R_{\tilde{\mathcal{L}}}(\boldsymbol{f})$, we have

$$\mathbb{P} \left\{ \sup_{\boldsymbol{f} \in \mathcal{F}} (\widehat{R}_{\tilde{\mathcal{L}}}(\boldsymbol{f}) - R_{\tilde{\mathcal{L}}}(\boldsymbol{f})) - \mathbb{E} \left[ \sup_{\boldsymbol{f} \in \mathcal{F}} (\widehat{R}_{\tilde{\mathcal{L}}}(\boldsymbol{f}) - R_{\tilde{\mathcal{L}}}(\boldsymbol{f})) \right] \ge \epsilon \right\} \le \exp \left( -\frac{2\epsilon^2}{n(\frac{K\pi\Theta}{n})^2} \right)$$

or equivalently, with probability at least $1 - \delta$,

$$\sup_{\boldsymbol{f} \in \mathcal{F}} (\widehat{R}_{\tilde{\mathcal{L}}}(\boldsymbol{f}) - R_{\tilde{\mathcal{L}}}(\boldsymbol{f})) \le \mathbb{E} \left[ \sup_{\boldsymbol{f} \in \mathcal{F}} (\widehat{R}_{\tilde{\mathcal{L}}}(\boldsymbol{f}) - R_{\tilde{\mathcal{L}}}(\boldsymbol{f})) \right] + K\pi\Theta \sqrt{\frac{\ln \frac{1}{\delta}}{2n}}$$

According to [Mohri et al., 2018], it is straightforward to show that

$$\mathbb{E} \left[ \sup_{\boldsymbol{f} \in \mathcal{F}} (\widehat{R}_{\tilde{\mathcal{L}}}(\boldsymbol{f}) - R_{\tilde{\mathcal{L}}}(\boldsymbol{f})) \right] \le 2\mathcal{R}_n(\tilde{\mathcal{L}} \circ \mathcal{F})$$

With the lemma 2, we complete the proof. $\square$

# 5 Proof of Theorem 3

Based on the Corollary 1, with $\boldsymbol{f}^* = \arg\min_{\boldsymbol{f} \in \mathcal{F}} R_{\mathcal{L}}(\boldsymbol{f})$, it is obvious to prove the generalization error bound as follows:

$$
\begin{aligned}
& R_{\mathcal{L}}(\hat{\boldsymbol{f}}) - R_{\mathcal{L}}(\boldsymbol{f}^*) \\
&= R_{\tilde{\mathcal{L}}}(\hat{\boldsymbol{f}}) - R_{\tilde{\mathcal{L}}}(\boldsymbol{f}^*) \\
&= \left( \widehat{R}_{\tilde{\mathcal{L}}}(\hat{\boldsymbol{f}}) - \widehat{R}_{\tilde{\mathcal{L}}}(\boldsymbol{f}^*) \right) + \left( R_{\tilde{\mathcal{L}}}(\hat{\boldsymbol{f}}) - \widehat{R}_{\tilde{\mathcal{L}}}(\hat{\boldsymbol{f}}) \right) + \left( \widehat{R}_{\tilde{\mathcal{L}}}(\boldsymbol{f}^*) - R_{\tilde{\mathcal{L}}}(\boldsymbol{f}^*) \right) \\
&\leq 0 + 2 \max_{\boldsymbol{f} \in \mathcal{F}} \left| \widehat{R}_{\tilde{\mathcal{L}}}(\boldsymbol{f}) - R_{\tilde{\mathcal{L}}}(\boldsymbol{f}) \right|
\end{aligned}
$$

The fist equality holds due to Corollary 1 and for the last line holds due to the fact $\widehat{R}_{\tilde{\mathcal{L}}}(\hat{\boldsymbol{f}}) \leq \widehat{R}_{\tilde{\mathcal{L}}}(\boldsymbol{f}^*)$ by the definition of $\hat{\boldsymbol{f}}$. Therefore, we obtain the first part of the theorem. $\qquad\square$

Before proving second part of the theorem, we first introduce some notations and provide the property of multi-label consistency.

The conditional risk of $\boldsymbol{f}$ can be defined as

$$
l(\boldsymbol{q}, \boldsymbol{f}) = \sum_{\boldsymbol{y} \in \mathcal{Y}} q_{\boldsymbol{y}} L(\boldsymbol{f}, \boldsymbol{y}),
$$

where $q_{\boldsymbol{y}} = (p(\boldsymbol{y}|\boldsymbol{x}))_{\boldsymbol{y} \in \mathcal{Y}}$ is a vector of conditional probability over $\boldsymbol{y} \in \mathcal{Y}$. Furthermore, we define the conditional $\mathcal{L}$-risk of $\boldsymbol{f}$ and the conditional Bayes $\mathcal{L}$-risk

$$
W(\boldsymbol{q}, \boldsymbol{f}) = \sum_{\boldsymbol{y} \in \mathcal{Y}} q_{\boldsymbol{y}} \mathcal{L}(\boldsymbol{f}, \boldsymbol{x}), \quad W^*(\boldsymbol{q}) = \inf_{\boldsymbol{f}} W(\boldsymbol{q}, \boldsymbol{f}).
$$

Based on the above notations, the definition of multi-label consistency can be formulated as follows.

**Definition 5.1.** *[Gao and Zhou, 2013] Given a below-bounded surrogate loss $\mathcal{L}$, where $\mathcal{L}(\cdot, \boldsymbol{y})$ is continuous for every $\boldsymbol{y} \in \mathcal{Y}$, $\mathcal{L}$ is said to be multi-label consistent w.r.t. the loss $L$ if it holds, for every $\boldsymbol{q}$, that*

$$
W^*(\boldsymbol{q}) < \inf_{\boldsymbol{f}} \{ W(\boldsymbol{q}, \boldsymbol{f}) : \boldsymbol{f} \notin \mathcal{A}(\boldsymbol{q}) \},
$$

*where $\mathcal{A}(\boldsymbol{q}) = \{ \boldsymbol{f} : l(\boldsymbol{q}, \boldsymbol{f}) = \inf_{\boldsymbol{f}'} l(\boldsymbol{q}, \boldsymbol{f}') \}$ is the set of Bayes decision functions.*

Based on the definition, the following theorem can be further established.

**Theorem 5.1.** *[Gao and Zhou, 2013] The surrogate loss $\mathcal{L}$ is multi-label consistent w.r.t. the loss $L$ if and only if it holds for any sequence $\{\boldsymbol{f}_n\}_{n \geq 1}$ that*

$$
\text{if} \quad R_{\mathcal{L}}(\boldsymbol{f}_n) \to R_{\mathcal{L}}^* \quad \text{then} \quad R(\boldsymbol{f}_n) \to R^*.
$$

which indicates the multi-label consistency is a necessary and sufficient condition for the convergence of $\mathcal{L}$-risk to the Bayes $\mathcal{L}$-risk, implying $R(\boldsymbol{f}) \to R^*$.

Now, if $\phi$ is a differential and non-increasing function with $\phi'(0) < 0$ and $\phi(t) + \phi(-t) = 2\phi(0)$, then Theorem 10 of [Gao and Zhou, 2013] tells us that there exists a non-negative concave function $\xi$ with $\xi(0) = 0$, such that,

$$
R(\boldsymbol{f}) - R^* \leq \xi(R_{\mathcal{L}}(\boldsymbol{f}) - \inf_{\boldsymbol{f}} R_{\mathcal{L}}(\boldsymbol{f})) \tag{10}
$$

which completes the proof. $\qquad\square$