# OpenReview forum: "Multi-Label Learning with Pairwise Relevance Ordering"
_NeurIPS.cc/2021/Conference — NeurIPS 2021 Poster_

### Official Review · Reviewer_guHb · 2021-07-13

**Rating:** 6
**Confidence:** 4

**Summary:**

This paper attempts to solve the multi-label learning problem with only pairwise relevance ordering. Authors argue that deciding the relative order of label pairs is obviously less laborious than collecting exact labels. A method is also presented to learn multi-label model with only pairwise relevance ordering and experiments shows its superiority.

**Ethical Concerns:**

No.

**Limitations And Societal Impact:**

Yes.

**Main Review:**

1. As stated in abstract, "deciding the relative order of label pairs is obviously less laborious than collecting exact labels", however, compared with the traditional exact labeling information where we only need to check whether each label belongs to the current instance, we need to annotate each instance many times when annotating it with relevance ordering labeling information (quadratic complexity w.r.t. the number of labels).  Therefore, it is necessary to discuss whether it is really less laborious for the proposed framework, or at least under what conditions, it will be less laborious. Author can also discuss the computational complexity (even the time costs) of the proposed method in Subsection 4.2. Besides, with exact labeling information, we can use binary relevance to learn the multi-label model, however, we can no longer learn such simple model with relevance ordering labeling information. Although binary relevance is usually criticized by not considering label correlations, it can achieve acceptable performance generally.

2. The related work can discuss more about learning with pairwise supervision, e.g.,
(a) H. Bao, et al. Classification from pairwise similarity and unlabeled data. In ICML, 2018.
(b) T. Shimada, et al. Classification from pairwise similarities/dissimilarities and unlabeled data via empirical risk minimization. Neural Computation, 2021.
(c) H. Bao, et al. Similarity-based Classification: Connecting Similarity Learning to Binary Classification. In arXiv, 2020.

3. As stated in the last sentence of Section 4, "the $\tilde{\pi}_{jk}^{10}$  can be directly estimated from the observed PRO training data". The estimated values might be affected by the number of label pairs annotated for each instance. Besides, why does the proposed method in Subsection 4.2 correspond to an unbiased estimator?

4. I cannot criticize that the theoretical analyses in Section 5 are useless, but authors should provide some more specific theoretical results rather than some less meaningful ones. For example, for one multi-label learning problem with a total of $q$ labels, how many label pairs are needed for the proposed method to achieve comparable performance with general multi-label classifiers (e.g., binary relevance) learned with exact labeling information? For another example, whether the proposed formulation is robust to noisy label pairs?

5. The experimental setup is unclear. For example, as stated in Subsection 6.1, "For each dataset, we randomly sample K pairs of labels and assign their relevance ordering", here, how to assign relevance ordering for a pair of labels with the same value (both relevant or irrelevant)?

6. It is better to provide some experimental results of well-established multi-label baselines to know the performance gap between learning with relative order of label pairs and learning with traditional exact labeling information.

7. It is better to annotate a real-world data set for the proposed framework rather than only generating some data sets from existing multi-label data sets. For example, authors can re-annotate the widely-used `scene' multi-label data set which only includes 2407 pictures and 6 labels. With the re-annotated data set, researchers can compare the performance/complexity of learners which are trained with different types of labeling information.

8. This paper is not carefully prepared. For example, Zhang & Zhou (2013) and Xu et al. (2019) are published in 2014 and 2020 respectively instead of 2013 and 2019. For another example, the style of reference citation is not in correct form throughout this paper.

**Time Spent Reviewing:**

12 hours

---

> ### Author Response · Authors · 2021-08-09
> **Thanks for your great efforts on the review of this paper. We will try our best to answer all your concerns.**
>
> Thanks for your great efforts on the review of this paper. We will try our best to answer all your concerns.
>
> Q1: It is necessary to discuss whether it is really less laborious for the proposed framework, or at least under what conditions, it will be less laborious.
>
> A1: On one hand, for annotators, picking one more relevant label out of two labels is relatively easier than picking multiple precise labels among whole label space, especially in some real-world scenarios, such as medical image analysis mentioned in the paper; on the other hand, even for a label space with medium size (contains dozens of labels), annotating precise labels often leads to some other issues, such as noisy labels and missing labels, etc. These problems are also weakly-supervised, and cannot be solved by standard multi-label methods, such as binary relevance.
>
> Q2: The related work can discuss more about learning with pairwise supervision.
>
> A2: In general, there exist some differences between learning with pairwise supervision (similarity learning for short) and PRO. Similarity learning focuses on binary classification and considers the pairwise similarity between two instances while PRO focuses on multi-label classification and considers pairwise relevance ordering between two labels. We would include detailed discussion about similarity learning and add citations of ICML’18, NC’21 and arXiv’20 in the revised version.
>
> Q3: The estimated values of $\tilde{\pi}^{10}_{jk}$ might be affected by the number of label pairs annotated for each instance.
>
> A3: In general, the larger number of the annotated label pairs, the more precise the estimated values. We would make our presentation clearer in the revised version.
>
> Q4: Why does the proposed method in Subsection 4.2 correspond to an unbiased estimator?
>
> A4: According to line 160 – line 161 in the paper, we show that $\mathcal{L}$-risk can be re-written as $R_{\tilde{\mathcal{L}}}(f)$ which can be estimated based on the PRO training examples.
>
> Q5: Authors should provide some more specific theoretical results rather than some less meaningful ones. … to achieve comparable performance with general multi-label classifiers (e.g., binary relevance) learned with exact labeling information …
>
> A5: In Section 5, instead of hamming loss (binary relevance in the review), we show that learning from PRO examples can be consistent with respective to ranking loss (another commonly used loss function in multi-label learning). That is as $n \rightarrow \infty$, $R_{\mathcal{L}}(\hat{f})=R^*_{\mathcal{L}} $, and then $ R(\hat{f}) = R^{*} $, i.e., the risk of $\hat{f}$ (trained on PRO examples) equals the Bayes risk with respective to ranking loss. The study on hamming loss is an interesting future direction. We also want to claim that consistency is important for multi-label learning, as previously studied in [1][2][3]. We do not think these theoretical results are less meaningful.
>
> [1] Convexity, classification, and risk bounds.
>
> [2] Consistent multilabel ranking through univariate loss minimization.
>
> [3] On the Consistency of Multi-Label Learning.
>
>
> Q6: Whether the proposed formulation is robust to noisy label pairs?
>
> A6: The proposed method is not designed specifically for handling noisy label pairs. The idea is an interesting future direction. In the current version, we focus on handling PRO examples in multi-label learning.
>
> Q7: How to assign relevance ordering for a pair of labels with the same value (both relevant or irrelevant)?
>
> A7: Thanks for your reminder. When two labels are both positive or negative, we decide their relevance ordering randomly, i.e., one out of two labels would be randomly chosen to be more relevant to the other one. We would make our presentation clearer in the revised version.
>
> Q8: It is better to provide some experimental results of well-established multi-label baselines to know the performance gap between learning with relative order of label pairs and learning with traditional exact labeling information.
>
> A8: Thanks for your suggestion. This is an interesting future work. In the current version, we focus on developing a new approach to allow multi-label learning with weakly supervised information.
>
> Q9: It is better to annotate a real-world data set for the proposed framework rather than only generating some data sets from existing multi-label data sets.
>
> A9: Thanks for your suggestion. We agree that it is important to construct a real dataset for the development of this new task. We would include these works in the future version.
>
> Q10: For example, Zhang & Zhou (2013) and Xu et al. (2019) are published in 2014 and 2020 respectively instead of 2013 and 2019. For another example, the style of reference citation is not in correct form throughout this paper.
>
> A10: Thanks for your reminder. After checking these citations carefully, we found both two papers have these two versions of references (Zhang & Zhou (2013, 2014) and Xu et al. (2019,2020)) in Google Scholar. We would carefully proofread and correct the format mistakes of references.

---

### Official Review · Reviewer_o4Co · 2021-07-16

**Rating:** 6
**Confidence:** 4

**Summary:**

In multi-label learning, the supervised information of pairwise relevance ordering is less informative than exact labels. Hence, it is an important challenge to effectively learn with such weak supervision. In this paper, a multi-label learning with pairwise relevance ordering approach is proposed. Meanwhile, it demonstrates that the unbiased estimator of classification risk can be derived with a cost-sensitive loss only from PRO examples. The theoretical analysis also has been provided.

**Limitations And Societal Impact:**


[1] The recent works are not fully explored. In the reference, the recent works are little. Meanwhile, more state-of-the-art methods should be compared.
[2] How about the time complexity of the proposed method with a large number of labels for each sample.
[3]The hinge loss is adopted in the paper. In the proposed method, it seems to be used as a ranking loss. It may be not clearly presented.


**Main Review:**

In this paper, a cost-sensitive loss function for learning a multi-label classifier with empirical risk minimization is proposed. To demonstrate the effectiveness of the proposed method, the theoretical analysis is provided to show that learning with PRO examples can be multi-label consistent to the commonly used ranking loss. The experimental results on multiple datasets and evaluation metrics demonstrate the practical usefulness of the proposed method.

**Time Spent Reviewing:**

1.5 hour

---

> ### Author Response · Authors · 2021-08-09
> **Thanks for your careful comments. We are glad to answer all your questions.**
>
> Thanks for your careful comments. We are glad to answer all your questions.
>
> Q1: The recent works are not fully explored. In the reference, the recent works are little.
>
> A1: Thanks for your suggestion. We would include more discussion about recent works in the revised version.
>
> Q2: More state-of-the-art methods should be compared
>
> A2: Since the proposed PRO is a novel learning framework, there is no method can be directly applied to PRO problems. To examine the effectiveness of the proposed method, in addition to the baseline methods, we also compare with a recently proposed noisy label learning method CCMN [1] by transforming the PRO problem into a MLL problem with class-conditional multi-label noise.
>
> Q3: How about the time complexity of the proposed method with a large number of labels for each sample?
>
> A3: The time complexity can be regarded as linear with respective to the number of label pairs, since we calculate the ranking loss for each label pair. Generally, as the number of label pairs increases, we would obtain more supervised information.
>
> Q4: The hinge loss is adopted in the paper. In the proposed method, it seems to be used as a ranking loss. It may be not clearly presented.
>
> A4: In the proposed method, the commonly used ranking loss is used as shown Eq.(1) in the paper. Since the ranking loss $L$ is highly discontinuous and computationally NP-hard, similar to previous works [2][3][4], we further use a surrogate loss $\mathcal{L}$ based on some surrogate loss functions $\phi$, such as hinge loss, etc.
>
> [1] CCMN: A General Framework for Learning with Class-Conditional Multi-Label Noise.
>
> [2] Minimizing the Misclassification Error Rate Using a Surrogate Convex Loss.
>
> [3] Log-Linear Models for Label Ranking.
>
> [4] A kernel method for multi-labelled classification.

---

### Official Review · Reviewer_YaKU · 2021-07-17

**Rating:** 8
**Confidence:** 4

**Summary:**

This paper studies the multi-label classification problem in a weakly supervised setting, where each instance is assigned with the relative order of label pairs. Authors formalize this task as a new framework called pairwise relevance ordering. An empirical estimator of the classification risk based on a cost-sensitive loss is proposed. Then authors theoretically prove that the proposed estimator is unbiased with the symmetric condition of the loss function. The error bound and consistency of the estimator is also proved. In addition to the theoretical results, the paper also validate the effectiveness of the proposed method empirically on multiple datasets.

**Ethics Review Area:**

["I don’t know"]

**Limitations And Societal Impact:**

no comment

**Main Review:**

The studied problem is interesting and novel. Weakly supervised information in the form of pairwise relevance orders is less costly to obtain compared to precise labeling. The study formalizes a reasonable and practical setting to allow multi-label learning with lower annotation cost.

For this new problem, authors propose a new estimator based on the cost-sensitive loss. The estimator is proved to be an unbiased estimator given a reasonable assumption, i.e., the loss function should be symmetric.

The paper made solid theoretical analysis both on the error bound and the consistency. The results show that learning with PRO examples can be multi-label consistent to the commonly used ranking loss.

Experiments are performed on 5 public datasets. Results on commonly used multi-label learning metrics show the superiority of the proposed method.

I think this paper presents a novel problem with effective solution. The contribution is significant from both theoretical and empirical aspects. Overall a solid work.

Below are some questions and suggestions:

It is not clear to me how the relevance ordering information obtained from the multi-label datasets. I guess it is obtained from the ground-truth labels. Authors stated that they randomly select pairs to assign the relevance order. Then how to decide the order in the case the two labels are both positive or negative? Authors need to make the details clear.

The proposed method shows a large superiority on VOC and MSCOCO datasets. While on MNIST datasets, u-CCMN performs quite well. Authors are suggested to give some detailed discussion on this phenomenon.

The consistency is proven with regard to the ranking loss. Can the results be generalized to other loss functions?

The references are not cited in the right form in the main content. It seems that authors mistakenly used \shortcite for all citations.


**Time Spent Reviewing:**

5 hours

---

> ### Author Response · Authors · 2021-08-09
> **Thanks for your appreciation of our paper. We are glad to answer all your questions.**
>
> Thanks for your appreciation of our paper. We are glad that you considered our work “novel, solid theoretical analysis, significant contribution, solid work”. We are glad to answer all your questions.
>
>
> Q1: How to decide the order in the case the two labels are both positive or negative? Authors need to make the details clear.
>
> A1: Thanks for your reminder. When two labels are both positive or negative, we decide their relative order randomly, i.e., one out of two labels would be randomly chosen to be more relevant to the other one. We would make our presentation clearer in the revised version.
>
> Q2: While on MNIST datasets, u-CCMN performs quite well. Authors are suggested to give some detailed discussion on this phenomenon.
>
> A2: Thanks for your suggestion. MNIST-style datasets can be regarded as easy multi-label classification tasks without complex label correlation. On one hand, u-CCMN transforms PRO task into multiple binary classification problem with noisy labels and utilize an unbiased estimator to solve each noisy label learning problem; on the other hand, the true noise rates are assumed to be known for u-CCMN, and thus providing some additional information for u-CCMN to boost the performance.
>
> Q3: The consistency is proven with regard to the ranking loss. Can the results be generalized to other loss functions?
>
> A3: The results would be difficult to generalized to other loss functions such as hamming loss, since the surrogate loss function is designed for ranking loss. However, it is possible to design an unbiased estimator with respective to other loss functions such as hamming loss and further prove its consistency. The study on other loss functions is indeed an excellent future direction.
>
> Q4: The references are not cited in the right form in the main content. It seems that authors mistakenly used \shortcite for all citations.
>
> A4: Thanks for your reminder. We would check carefully and correct the form of references.

---

### Official Review · Reviewer_d3TV · 2021-07-20

**Rating:** 8
**Confidence:** 4

**Summary:**

This paper presents a new multi-label learning framework called Pairwise Relevance Ordering (PRO). In this setting, the supervised information is given in the form of relevance orders between labels pairs.

To solve the PRO problems, an unbiased estimator of the classification risk is proposed, whose properties of unbiased, consistency and error bound are theoretically proven.

Experiments are performed on benchmark datasets to show the superiority of the proposed method.


**Limitations And Societal Impact:**

No potential negative social impact.

**Main Review:**

The paper presents a new framework of PRO for multi-label learning. Because it does not require annotators to precisely assign class labels for each instance, it is expected to be more cost-effective.

The authors propose an unbiased estimator of classification risk, which is shown to be derived with a cost-sensitive loss only from PRO examples. This allows to perform effective multi-label learning with weakly supervised information.

The authors give in-depth theoretical analysis to derive the error bound. They also prove that the estimator is consistent with the commonly used ranking loss.

Extensive experimental results on multiple datasets and evaluation metrics demonstrate the practical usefulness of the proposed method.

There are some suggestions:
The experimental settings are less clear. For example, it is not introduced how to assign relevance ordering information to label pairs given that the standard multi-label datasets are used in the experiments.
There are some language mistakes. Careful proofreading is needed. Especially, the reference citation is not in the correct form.

**Time Spent Reviewing:**

4

---

> ### Author Response · Authors · 2021-08-09
> **Thanks for your appreciation of our paper. We are glad to answer all your questions.**
>
> Thanks for your appreciation of our paper. We are glad to answer all your questions.
>
> Q1: … how to assign relevance ordering information to label pairs given that the standard multi-label datasets are used in the experiments?
>
> A1: Thanks for your suggestion. As mentioned in Datasets of Section 6.1 (line 212-215), for each dataset, we randomly sample K pairs of labels and assign their relevance ordering based on their ground-truth labels. We will make our presentation clearer in the revised version. When two labels are both positive or negative, we decide their relative order randomly, i.e., one out of two labels would be randomly chosen to be more relevant to the other one. We would make our presentation clearer in the revised version.
>
> Q2: There are some language mistakes. Careful proofreading is needed. Especially, the reference citation is not in the correct form.
>
> A2: We would carefully proofread and correct the language mistakes as well as the form of references. Thanks a lot

---

> ### Comment · Reviewer_d3TV · 2021-08-13
> **I read the rebuttal from authors and satisfied with their clarification and efforts.**
>
> I read the rebuttal from authors and satisfied with their clarification and efforts.

---

### Decision · Program_Chairs · 2021-09-27

**Decision:**

Accept (Poster)

**Comment:**

The paper presents a new framework for multi-label learning that does not require annotators to precisely assign class labels for each instance and thus is cost-effective. The theory part supports the algorithm design and the experiments show good results. The author's responses addressed the questions of the reviewers.